# Diversity and community distribution of soil bacterial in the Yellow River irrigation area of Ningxia, China

**Xia Wu, Jinjun Cai** *****, **Zhangjun Wang, Weiqian Li, Gang Chen, Yangyang Bai**

Institute of Agricultural Resources and Environment, Ningxia Academy of Agriculture and Forestry Science, Yinchuan, China

***** nxyccai@163.com

## Abstract

The bacterial community performs an essential ecological role in maintaining agriculture systems. The roles of bacteria in the forest, marine, and agricultural systems have been studied extensively and intensively. However, similar studies in the areas irrigated by the Yellow River remain limited. In this study, we used Illumina sequencing analysis with the 16S rRNA method to analyze the bacterial diversity, community structure, and influencing factors in soil samples from eight regions of the Yellow River irrigation area in northwestern China. The bacterial community structure and diversity varied among samples from the eight regions. The samples differed significantly in terms of the bacterial community composition. Proteobacteria (approximately 12.4%-55.7%) accounted for the largest proportion and was the dominant bacteria, followed by Actinobacteria (approximately 9.2%-39.7%), Bacteroidetes (approximately 1.8%-21.5%), and Chloroflexi (approximately 2.7%-12.6%). Among the physicochemical variables, the soil pH in the eight regions was mildly alkaline, and the total nitrogen, total phosphorus, and total potassium contents in the soils differed significantly. However, the trend in the variations of the above variables was essentially similar. Soil bacteria in Yongning county had greater Chao1, Shannon-Wiener, and Simpson indices than those in the other regions. Notably, soil moisture, organic matter, and total nitrogen were recognized as the primary factors influencing the bacterial community in the Yellow River irrigation area. Our results revealed the laws of variation in soil bacterial diversity and community composition in the Yellow River irrigation area. Our findings could be beneficial for maintaining sustainable ecological practices in the Yellow River irrigation area.

## Introduction

The global population is increasing and has now reached 7 billion [1]. Experts speculate that the global population will reach 9.5 billion by 2050, which will put enormous pressure on food supply worldwide. FAO (2013) reported that the food deficit will be 70% in the coming decades [2, 3]. To address this problem, soils are planted with successive crops, which depletes nutrient reserves, leads to a negative nutrient balance, and causes soil degradation. Soil

**Funding:** The authors wish to express their sincere thanks to the Ningxia Hui Autonomous Region Sci-Tech Innovation Demonstration Program of High-Quality Agricultural Development and Ecological Conservation (NGSB-2021-11-01),and the Ningxia Hui Autonomous Region Key R&D Project (2023BEG02042) jointly provide funding support for research work, including soil sampling investigation, soil sample testing, data collection and analysis, decision to publish, preparation of the manuscript. The authors are grateful to thank Guangzhou Genedenovo Biotechnology Co., Ltd. for testing and assisting in microbiome analysis.

**Competing interests:** The authors have declared that no competing interests exist.

microorganisms are the drivers of material transformation and nutrient cycling in soil, which are crucial for carbon storage, plant growth, and the protection of other organisms, especially from pathogens [4–7]. Numerous studies have demonstrated the critical role of soil microbial diversity in soil functions, including the carbon-nitrogen cycle, pathogen management, and bioremediation. Specific soil microbial groups can restrain soil-borne plant pathogens [8–11]. As the most diverse and abundant group of soil microorganisms, bacteria are ubiquitous owing to their ability to evolve and survive in all types of environments [12–14]. Soil bacteria are involved in most soil ecological processes, such as geochemical cycles, energy flow, and information transfer [15–17]. Soil bacteria are sensitive to environmental changes, and their community composition and diversity respond rapidly to changes in environmental conditions, making them an important indicator for assessing the quality of the soil environment [18–21]. A good soil bacterial structure and higher bacterial diversity can increase bacterial activity and improve the physicochemical properties of soil and the contribution to soil nutrient cycling. Therefore, understanding the changes in soil bacterial community composition under different environmental conditions systematically is a necessity.

In agroecosystems, soil bacteria are a major class of microorganisms that keep soils healthy and productive [22]. Soil bacteria perform important functions in the soil, breaking down organic residues from enzymes released in the soil. Many bacteria secrete enzymes in the soil to increase the solubility and bioavailability of phosphorus. Most soil bacteria perform better in well-oxygenated soils with a neutral pH. Nitrogen is often lacking in soils. Bacteria provide nitrogen to plants in large quantities. However, some studies have shown that soil bacterial communities are profoundly influenced by changes in soil conditions. For example, soil pH has been considered to influence bacterial community composition [23, 24] and has also been used for the large-scale prediction of bacterial diversity [25, 26]. Changes in the soil nitrogen concentration exert a significant effect on the structure and composition of the bacterial community [27]. In addition, soil moisture, plant diversity, and soil type have been shown to be associated with changes in bacterial communities [28, 29]. Importantly, there is ample evidence that bacterial communities respond directly or indirectly to soil environmental changes. In other words, fertilization, changes in pH, and changes in farming patterns have been shown to affect soil bacterial communities. Thus, a deeper understanding of how bacterial communities respond to soil conditions is important for evaluating agroecosystem processes.

Yinchuan Plain, commonly referred to as Ningxia Plain, is the political, economic, and cultural center of the Ningxia Province. Situated in the middle of Ningxia Province on the two banks of the Yellow River, it comprises a typical alluvial plain formed by long-term siltation from the Yellow River [30]. It is an important crop production area in Ningxia as well as the primary irrigation area of the Yellow River, which is crucial for preserving ecological security in northwest China and has a history of more than 2000 years [31]. Unfortunately, the physicochemical and microbiological properties of the soil have fundamentally changed owing to long-term planting, irrigation, fertilization, and crop planting, adversely affecting the development of the ecological environment and agriculture. Therefore, research on soil bacteria can help us better understand the soil properties and soil quality conditions in the irrigation area.

To gain insights into soil biochemical processes in irrigation areas, we used soil samples from a typical irrigated area in the Ningxia Plain and examined soil bacterial changes and diversity. Our specific objectives were (i) elucidating changes in the composition and diversity of the soil bacterial community in the irrigated area and (ii) defining the relationship between soil bacterial communities and soil physicochemical properties.

## Materials and methods

### Study area

The study area is situated in Yinchuan Plain (37˚46'~39˚23'N, 105˚45'~106˚56'E), with a surface area of 7615 km$^2$, spanning approximately 165 km from north to south and 60 km from east to west. Although the Yinchuan Plain lies in the arid inland temperate zone, with an average annual precipitation of less than 200 mm, average annual temperature of 8.3˚C, and annual evaporation of nearly 2000 mm, the Yellow River flows through obliquely with a flow of approximately 280 km, carrying around 32.5 billion m$^3$ of transit water annually [30]. Moreover, the Yinchuan Plain has annual sunshine of up to 3000 h, with abundant agricultural natural resources such as light, heat, water, and soil. These factors make it an important irrigated agricultural area in northwest China. Corn, wheat, rice, fruits, vegetables, and other crops produced in the Ningxia Plain have high and stable yields and good quality.

### Soil collection

From July to October of 2020 to 2021, we collected 147 soil samples from eight areas irrigated by the Yellow River across the Yinchuan Plain, including Huinong District, Pingluo County, Yinchuan City, Helan County, Yongning County, Lingwu City, Litong District, and Qingtongxia City. Representative fields were chosen for each irrigated site based on the field area, soil type, and crop planting conditions. In each sample site, samples were collected at 10~15 points using the S-shaped or the quincunx sampling method at 0 to 20 cm depth. Three plot replicates were established at each site. Following this, the apparent root system was removed, and 2 kg of the mixed soil samples was reserved using the quartering method. Next, 100 g and 400 g of soil samples were stored in the refrigerator. After the samples were brought back to the laboratory, 100 g of the soil samples were stored at -80˚C in a refrigerator for the soil bacterial diversity test, and 400 g of the samples were stored at 4˚C in a refrigerator for testing the content of ammonium nitrogen and nitrate nitrogen in soil. The remaining 1.5 kg of samples were stored in a large valve bag for examining the physicochemical properties after air drying and screening. Geographic information, including the latitude, longitude, and altitude for each sample site was recorded by GPS (GM977-GPS, Shanghai, China), and the specific information of each sampling region is listed in Table 1.

### Soil physicochemical properties

The physical properties of the soil samples were assessed using soil argrochemistry analysis [32]. Soil pH was measured using a pH meter (soil: water = 1:5 (w:v)) (Mettler FE28-Standard desktop, Shanghai, China). Soil moisture was measured using the Sartorius MA 100 moisture test apparatus. Soil bulk density and field capacity were measured using the cutting ring method. The total salt content was measured using a conductivity meter (INESA DDS-307, USA). Soil organic matter was measured using potassium dichromate ($K_2Cr_2O_7$). Total nitrogen was measured using an automatic azotometer (GL-500, Shandong, China). Total and available phosphorus were measured using the molybdenum-antimony anticolorimetric method. A flame photometer for used for measuring total and available potassium [33]. $NH_4^+$-N and $NO_3^-$-N were assayed using an ultraviolet spectrophotometer [34].

### Illumina sequencing analysis of 16S rRNA gene amplicons

Soil DNA was extracted from all 147 samples using HiPure Soil DNA Kits (or HiPure Stool DNA Kits) (Magen, Guangzhou, China) according to the manufacturer's protocol. A Nano-Drop Micro Spectrophotometer (NanoDrop 2000, Thermo Fisher Technology, USA) and

**Table 1. Characteristics of the study areas.**

| Sampling Area | Sample Size | Farmland Area (10000 hectares) | Average Temperature (˚C) | Rainfall (mm) | Sunshine Hours(h) | Soil Type | Crops |
|---|---|---|---|---|---|---|---|
| Huinong District (HN) | 15 | 2.81 | 10.7 | 137.5 | 3715.7 | Irrigation-silting soil, fluvo-aquic soil, aeolian sandy soil | Corn, oil sunflower, alfalfa |
| Pingluo County (PL) | 47 | 6.38 | 9.7 | 141.1 | 3680.7 | Irrigation-silting soil, fluvo-aquic soil, solonchak, aeolian sandy soil | Rice, corn, wheat, vegetables |
| Helan County (HL) | 18 | 4.28 | 9.9 | 147.0 | 3486.9 | Irrigation-silting soil, fluvo-aquic soil, solonchak, aeolian sandy soil | Rice, corn, vegetables |
| Yinchuan City (YC) | 9 | 3.88 | 10.8 | 145.5 | 3487.5 | irrigation-silting Soil, sierozem, aeolian sandy soil | Rice, corn, vegetables |
| Lingwu City (LW) | 11 | 2.44 | 9.8 | 149.8 | 3554.5 | Irrigation-silting soil, fluvo-aquic soil | Rice, corn |
| Yongning County (YN) | 6 | 3.53 | 11.1 | 177.2 | 3671.1 | Irrigation-silting soil, sierozem | Rice, corn, vegetables |
| Litong District (LTQ) | 10 | 3.08 | 11.2 | 205.0 | 3444.8 | Irrigation-silting soil, fluvo-aquic soil | Rice, corn, vegetables |
| Qingtongxia City (QTX) | 31 | 3.91 | 10.1 | 200.9 | 3348.1 | Irrigation-silting soil, fluvo-aquic soil, sierozem, alluvial soil | Rice, corn, wheat, vegetables |

agarose gel electrophoresis were used to determine the concentration, purity, and integrity of DNA. PCR of the V3-V4 region of 16S rDNA gene was performed using primers 341F (5'-CCT ACG GGN GGC WGC AG-3') and 806R (5'-GGACTACHVGGGTATCTAAT-3') [35] to study the bacterial communities. For PCR, we used 5 μL of 10× KOD Buffer, 5 μL of 2.5 mM dNTPs, 1.5 μL of each primer (5 μM), 1 μL of KOD Polymerase, and 100 ng of template DNA (Guangzhou, China). The representative sequences were classified into organisms by a naive Bayesian model using the RDP Classifier based on SILVA Database (version 132),with the confidence threshold value of 0.8.

## Statistical analysis

The correlation among the abundance of soil bacterial communities, environmental factors, and the composition of bacterial species was analyzed on the platform by Guangzhou Gene Denovo Biotechnology Co., Ltd (Guangzhou, China). Data were analyzed using R (4.3.1). Chao1 [36] and Shannon indexes [37] were calculated for alpha diversity (Caporaso & Gregory, 2010). Non-metric multi-dimensional scaling (NMDS) of UniFrac distances was conducted, and the results were plotted in R (4.3.1). Redundancy analysis (RDA) was used to visualize the relationship between microbial communities and environmental variables. The differences in bacterial communities between eight regions were assessed using one-way ANOVA, followed by Tukey's honestly significant difference post hoc test. Statistical analyses were performed using SPSS25.0 (IBM, Armonk, NY, USA) and the vegan package in R(4.3.1).

## Results

### Physicochemical properties of soil samples

Various physicochemical variables were used to characterize the soil environment. Among the different soil physicochemical variables (Fig 1), the pH values of the eight regions were weakly alkaline (Fig 1A and 1B), and the total nitrogen, total phosphorus, and total potassium contents in the soil differed significantly. However, the trend in variation was essentially the same. The highest total nitrogen was observed in soil from HN, and the lowest was observed in soil from HL. PL had the highest soil moisture content and field capacity, followed by HN and QTX, and YC had the

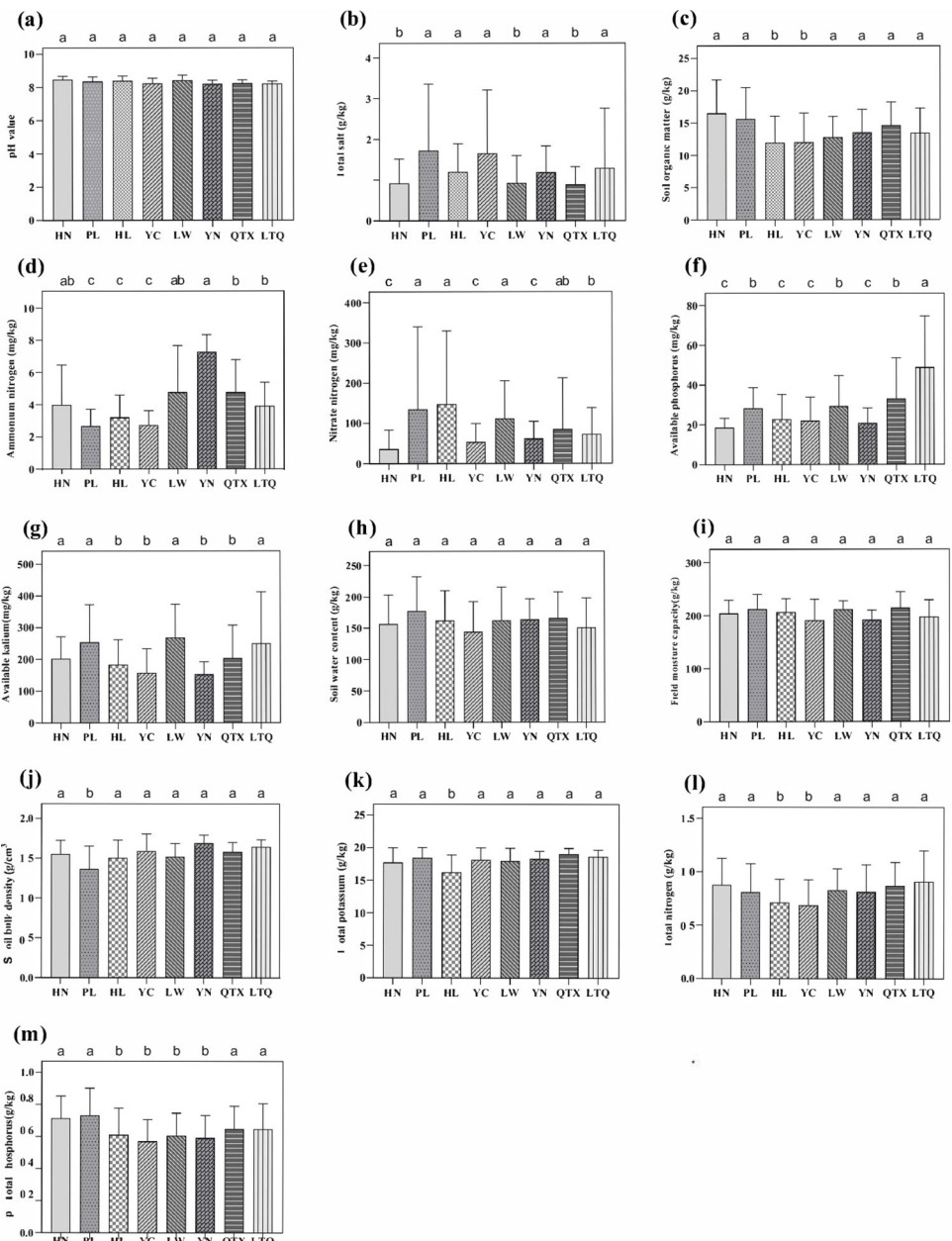

**Fig 1. Variation of soil physicochemical properties.** Note: HN: Huinong District; PL: Pingluo County; HL: Helan County; QTX: Qingtongxia City; LTQ: Litong District; YN: Yongning County. Similarly, hereinafter.

lowest. The ammonium nitrogen content was highest in soil from YN and lowest in soil from YC, and the nitrate nitrogen content was highest in soil from PL and HL and lowest in soil from HN. The available phosphorus and potassium contents were highest in LTQ and lowest in YC and YN. The soil organic matter content was highest in HN and lowest in YC and HL.

## Bacterial community structure

We classified the top 10 operational taxonomic units (OTUs) bacteria among soil bacterial communities in different regions based on the phylum and class to identify the dominant

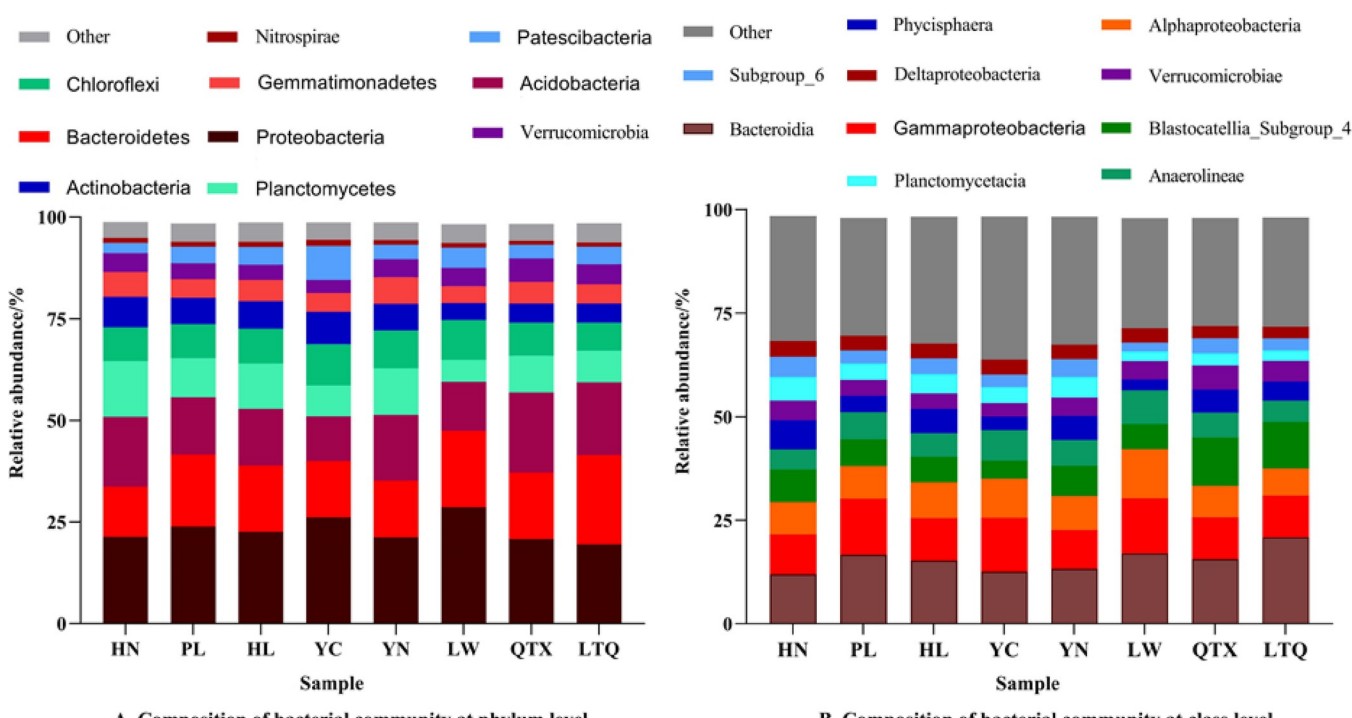

**Fig 2. Composition of soil bacterial community in different regions.** Note: HN: Huinong District; PL: Pingluo County; HL: Helan County; QTX: Qingtongxia City; LTQ: Litong District; YN: Yongning County.

bacterial communities in different fields (Fig 2). In all soil samples obtained from Yellow River irrigation areas, at the phylum level, bacterial profiles were dominated by Proteobacteria (23%), Bacteroidetes (16.4%), Acidobacteriota (15.2%), Planctomycetes (9.4%), Chloroflexi (9.8%), Actinobacteria (8.6%), Gemmatimonadetes (6.1%), Verrucomicrobia (4.6%), Patescibacteria (4.9%), and Nitrospirae (1.5%). At the class level, Bacteroidetes (20.87%) were predominant, followed by Blastocatellia_Subgroup_4 (11.31%), Gammaproteobacteria (10.09%), Alphaproteobacteria (6.50%), and Anaerolineae (5.10%); other classes had an abundance less than 5% (Fig 2A).

Among the eight areas, LW had the highest relative abundance of Proteobacteria (28.64%), followed by YC (26.15%), PL (23.84%), HL (22.61%), HN (21.30%), YN (21.16%), QTX (20.76%), and LTQ (19.52%). In the LW sample site, the relative abundance of Bacteroidetes (18.85%) was the second-highest, followed by the abundance of Acidobacteriota (11.98%), Chloroflexi (9.87), Planctomycetes (5.41%), Patescibacteria (4.99%), Gemmatimonadetes (4.22%), Verrucomicrobia (4.44%), Actinobacteria (4.09%), and Nitrospirae (1.21%). Overall, the relative abundance of Verrucomicrobia and Acidobacteriota in QTX was the highest, that of Gemmatimonadetes in YN was the highest, and those of Actinobacteria, Patescibacteria, and Nitrospirae in YC were the highest. At the class level, the relative abundance of Bacteroidetes was the highest in LTQ (20.87%), followed by that in LW (16.98%), PL (16.64%), QTX (15.59%), HL (15.28%), YN (13.26%), YC (12.60), and HN (11.91%). Gammaproteobacteria was the second-most abundant, ranging from 13.25% in LW to 9.63% in HN. In general, LTQ had the highest relative abundance of Bacteroidetes (20.87%). QTX had the highest relative abundance of Blastocatellia_Subgroup_4. LW had the highest relative abundance of Alphaproteobacteria. YC had the highest relative abundance of Anaerolineae. PL had the highest relative

abundance of Gammaproteobacteria. HN had the highest relative abundances of Phycisphaera, Deltaproteobacteria, Verrucomicrobia, Deltaproteobacteria, and Subgroup_6 (Fig 2B).

## Soil bacterial diversity

The soil bacterial diversity varied significantly across the eight sample sites, with YN and LW having higher Chao1 indexes than other sample plots (Fig 3A). The ACE index of YN was significantly higher than that of other sample; however, the values did not differ significantly (Fig 3B). The Shannon indexes of soil bacteria in HL and YN were higher than those in HN, YC, LW, QTX, and LTQ (Fig 3C). The Good'scoverage index, which reflects the sequencing depth, indicated that the similarity of all samples in this study was more than 97%,"showing homogeneous sequencing coverage across samples. Therefore, the findings can accurately represent the community characteristics of soil bacteria in different analysis and indicate the actual bacterial distribution of each sample (Fig 3D).

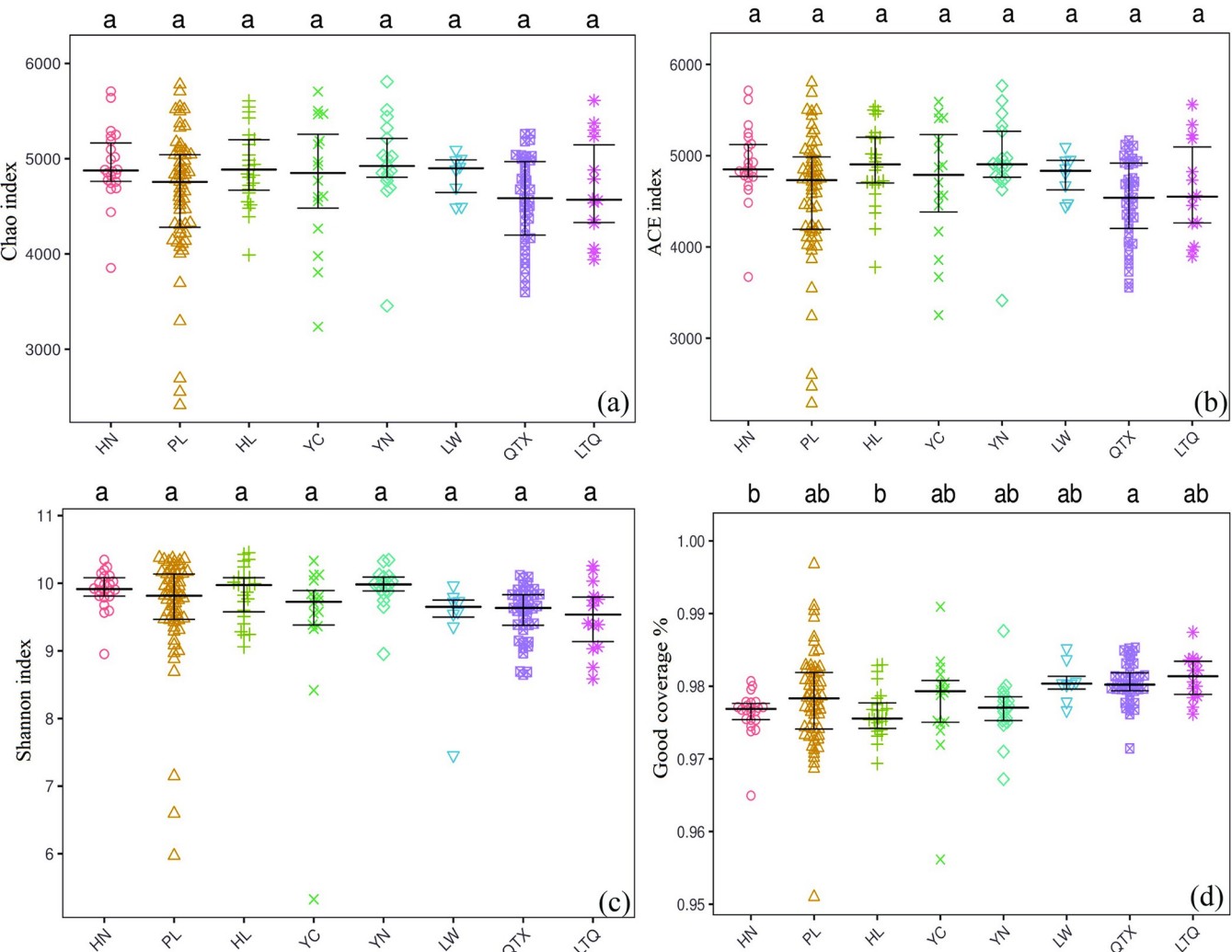

**Fig 3. α diversity comparison of soil bacteria in different regions.** (a): Chao index; (b): ACE index; (c): Shannon index; (d): Good coverage. Note: HN: Huinong District; PL: Pingluo County; HL: Helan County; QTX: Qingtongxia City; LTQ: Litong District; YN: Yongning County.

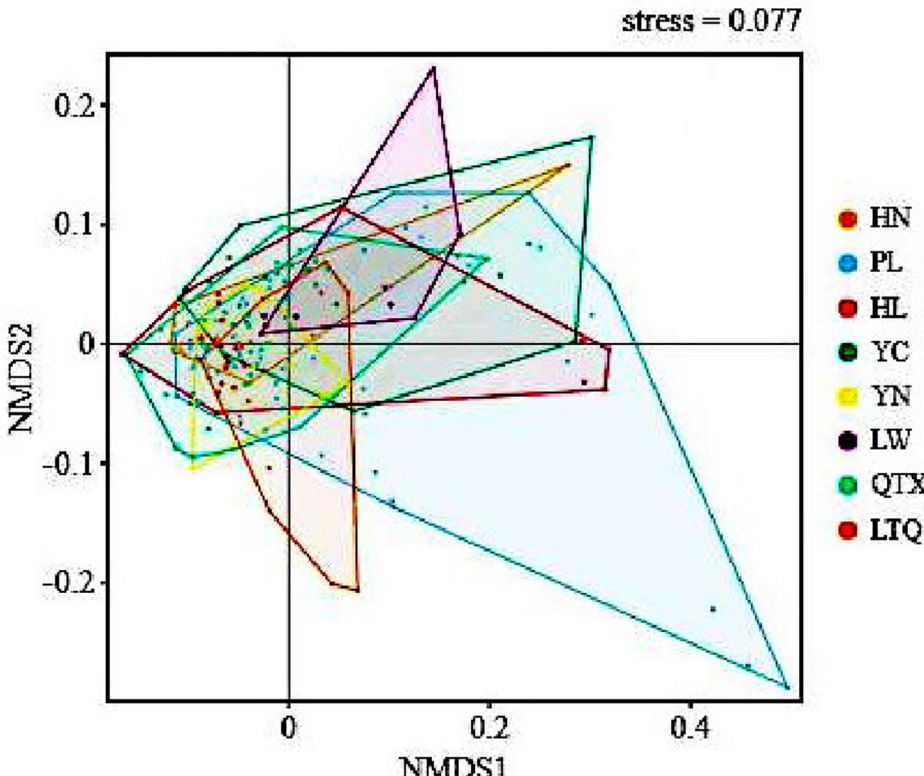

**Fig 4. Non-metric multidimensional scaling (NMDS)-based similarity analysis of soil bacterial community in different regions.** Note: HN: Huinong District; PL: Pingluo County; HL: Helan County; QTX: Qingtongxia City; LTQ: Litong District; YN: Yongning County. Similarly, hereinafter.

The similarity in the soil bacterial community structures in the Ningxia Plain was examined in this study based on the non-metric multidimensional scaling analysis (NMDS) method. The stress value of NMDS was 0.113 (<0.2), indicating that the analytical results precisely reflected the composition differences in soil bacterial communities in the different farmlands in Ningxia (Fig 4). The result indicated significant differences in the soil bacterium types and bacterial community composition across different regions ($R^2$ = 0.13, p<0.001). The sample points between PL and YC showed significant differences and minor similarities. The differences in soil samples between the LTQ and QTX regions were minor, and the bacterial communities were largely similar.

## Relation between soil physicochemical properties and soil bacteria

We analyzed the relationship between bacterial community composition and edaphic factors through redundancy analysis. As shown in Fig 5, collectively, the cumulative squared Pearson's correlation coefficient of the first two axes was 85.14%, which can better depict the relation between dominant bacteria and edaphic factors. The soil moisture content, organic matter, and total nitrogen had a significant impact on the soil bacterial community (Fig 6). Additionally, at the phylum level, the soil moisture content was positively correlated with Patescibacteria and was negatively correlated with Planctomycetes, Gemmatimonadetes, and Acidobacteriota, the soil organic matter content and pH value were positively correlated with

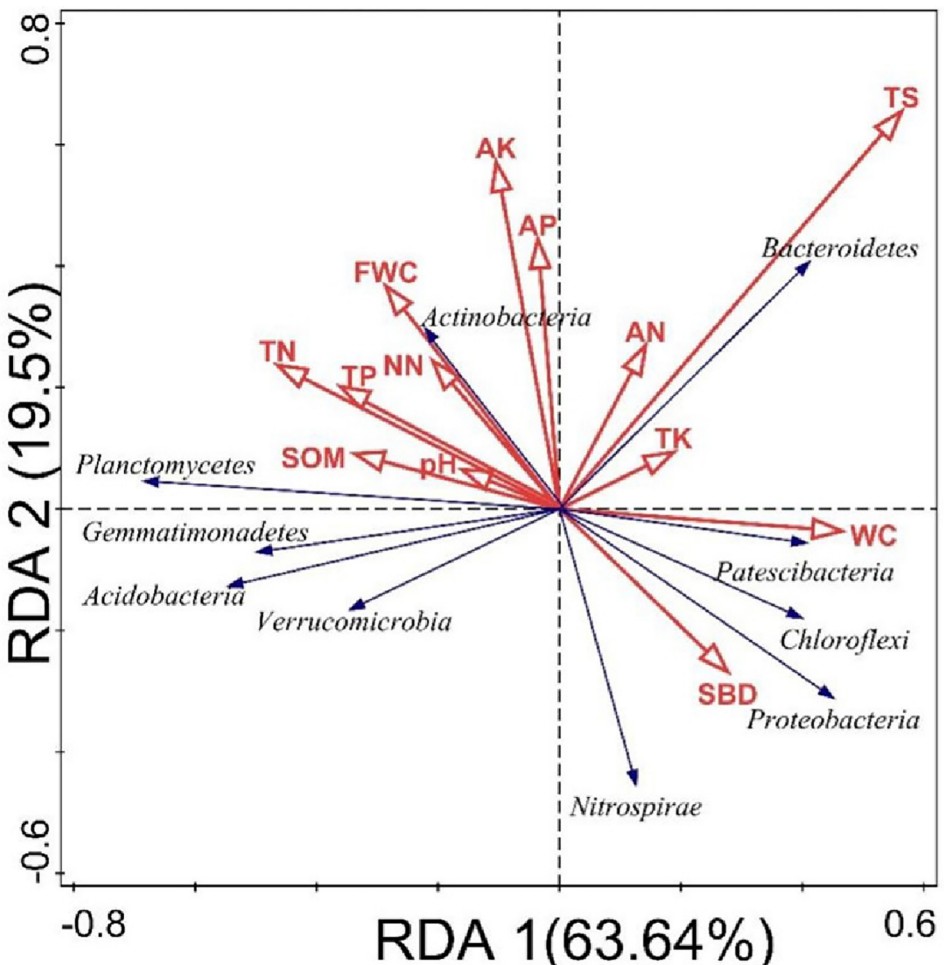

**Fig 5. Correlation analysis of bacterial community composition and edaphic factors at phylum level.**

the abundance of Planctomycetes, the total salt content was positively correlated with the abundance of Bacteroidetes.

## Discussion

Soil bacteria are a major group of microorganisms that keep soils healthy and productive. In our study, we identified the bacterial community in the Yellow River irrigation area using the microbial diversity analysis and found that the moisture, organic matter, and total nitrogen contents were the primary factors influencing the bacterial community in the Yellow River irrigation area in Ningxia, China. The findings of this study provide strong evidence for the evaluation and management of soil quality in the Yellow River irrigation area in Ningxia province.

### Soil physicochemical properties

Soil and water quality plays an important role in the Yellow River irrigation area. A better understanding of soil and water quality will help irrigators efficiently manage their crops [38]. In this study, the eight regions had a high pH, which may have negatively affected nutrient absorption from the soil. Additionally, soil properties, especially ammonium nitrogen, nitrate

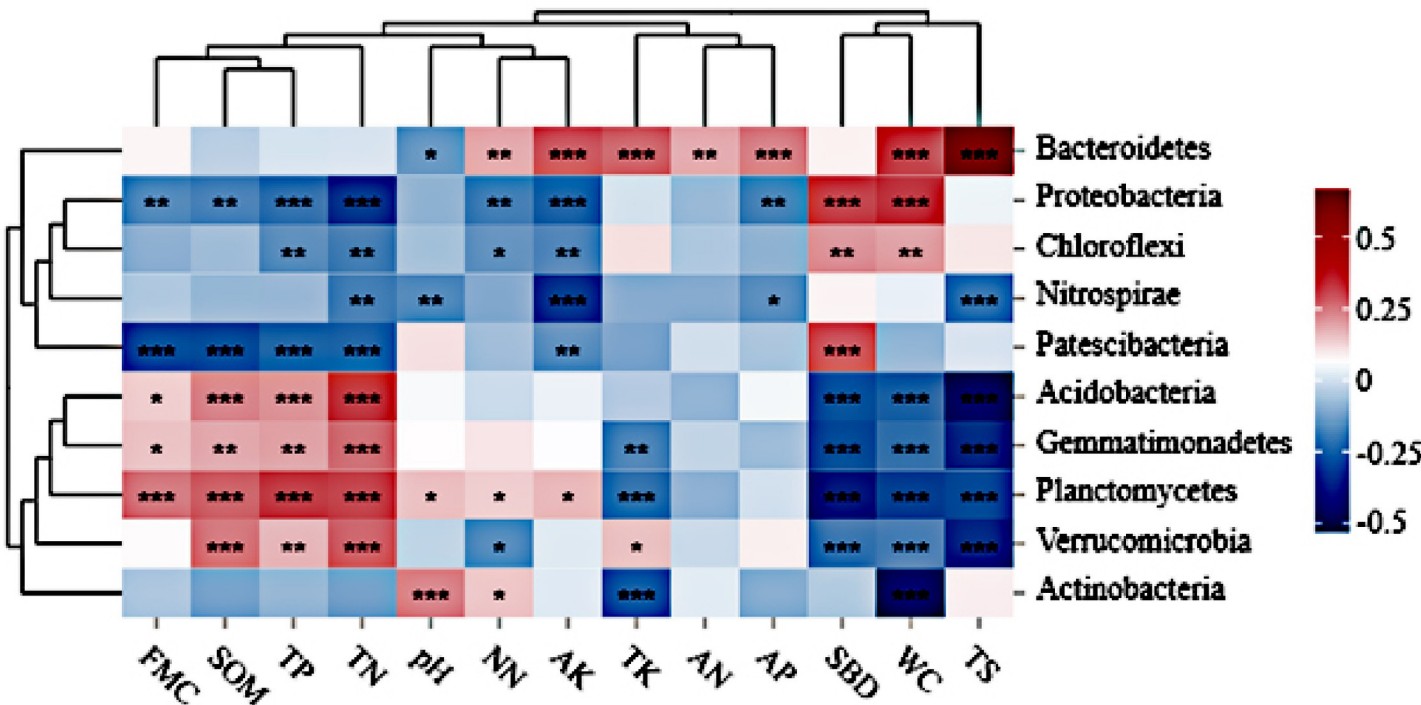

**Fig 6. Heatmap of cluster analysis of the top 10 phyla in different regions.** Note: TN is total nitrogen; TS is total salt; TP is total phosphorus; TK is total potassium; SBD is soil bulk density; WC is soil water content; pH is pH value; SOM is soil organic matter; NN is nitrate nitrogen; FMC is field moisture capacity; AK is soil available kalium; AP is available phosphorus; AN is ammonium nitrogen.

nitrogen, total phosphorus, and available phosphorus levels, were significantly different among the sample sites. These may result in the abundance of bacterial taxa in different sample sites and also indicate that bacterial communities have a relatively stable survival environment to ensure function.

## Bacteria response to yellow river irrigation area

The composition of the soil bacterial community can reflect changes in soil quality [39, 40]. In our study, despite the obvious differences in the spatial scale and environmental conditions among the eight regions, some conditions are relatively similar. For example, the soil in the eight regions had a high moisture and total nitrogen content. These may result in an abundance of bacterial diversity and relatively stable compositions. At the phylum level, Proteobacteria (23%), Bacteroidetes (16.4%), and Acidobacteriota (15.2%) were the dominant groups, with a relatively high abundance in the study area, which is consistent with findings from previous studies. This indicated an absolute superiority of Proteobacteria in the bacterial community structure in farmland soil [41–43]. Proteobacteria, as the most important bacterial group and one of the most abundant soil microorganisms, has been shown to be a dominant community in farmland [44] and mining areas [45, 46]. Its metabolic activity is the most prominent bacterial activity in the soil. At the class level, the dominant bacteria (Gemmatimonadetes and Alphaproteobacteria) are Proteobacteria as well, which emphasizes the essential role of Proteobacteria in the bacterial community structure of farmland soils. Generally, members of Proteobacteria are present widely in various types of soil [47]. Therefore, we confirmed that certain bacterial taxa are not unique to the Yellow River irrigation area, and their relative abundance values are simply different. A higher relative abundance of Proteobacteria is observed in LW

and YN, followed by that in YC, with a lower abundance in LTQ and QTX. Previous findings on Proteobacteria (which can be used as indicators of the nutrient content) indicated that increasing Proteobacteria abundance may be beneficial for improving crop yield [48–50]. The higher Proteobacteria abundance in LW and YN in our study could be beneficial for the improvement of the soil nutrient content and yield.

Soil bacteria are the principal microorganisms that keep soil healthy and productive. Soil bacterial diversity is critical to the integrity of soil ecosystems and long-term soil sustainability [41, 51]. The difference in bacterial community structures is also considered to influence the diversity of soil bacteria; the greater the diversity of the soil bacterial community, the more stable the soil ecosystem. In this study, the Shannon indexes of soil bacteria in HL and YN were higher than those in other regions. Interestingly, the Chao1, Shannon, and ACE indexes are similar in the eight regions, even though the physical and chemical properties of the soil, such as the total nitrogen, total phosphorus, and available phosphorus levels were significantly different. This may be attributed to the long-term application of fertilizers and the presence of soil nutrients that have a strong filtration effect on soil bacteria [37, 38]. Soil bacteria are reportedly more sensitive to N and P enrichment; for example, changes in $NH_4^+$-N and $NO_3^-$N content can explain the effects of the bacterial community [39]. Moreover, soil bacterial diversity is primarily controlled by factors such as pH. The soil samples used in this study were alkaline (pH > 7.0); this may be resulting the less influence of pH on bacterial communities in the Yellow River irrigation area. Meanwhile, the heterogeneity of the physical and chemical properties of soil samples is also an important factor affecting soil bacterial diversity [52, 53]. The plausible explanation is that the eight regions have similar soil types and soil pH, because of which the pH value considerably exceeds the suitable range (<5.5) for bacterial growth. Thus, the soil properties of the Yellow River irrigation area are in a relatively balanced state, and the difference in soil bacterial diversity in the eight regions could be used as an important reference for the management of soil quality in the Yellow River irrigation area.

In addition, some studies have revealed that β diversity can also indicate differences in soil bacterial composition among agriculture systems [54]. Also, some plants are known to affect bacterial communities in the soil, and the diversity of such plants is often used to predict the β diversity of soil bacterial communities [40, 41]. In our study, the composition of bacterial communities at the phyla level differed in the PL and YC regions, whereas the LTQ and QTX regions showed similarities in bacterial communities. This could be attributed to the proximity of the latter two regions and the similarity in soil conditions. The sample points in PL and YC showed significant differences and minor similarities, which could be attributed to similar soil conditions and crops [49]. The crops in PL were dominated by wheat, apart from rice and corn; this could explain the variations in the bacterial communities in PL and YC. Previous studies have shown that soil bacteria can aggregate in response to different environmental factors [55], which is not consistent with our conclusions. This could be attributed to the selection of different variables in the study. Although we could not determine the factors influencing the difference in diversity, this difference could help us understand the variations in soil parameters in the different regions in the irrigation area.

## Abundance of bacterial communities based on the environmental variables

For the sustainable production of field crops, food security, and economic prosperity, healthy and productive soil is a necessity [56, 57], and nutrient availability and soil quality preservation are both considerably influenced by soil microorganisms [49, 58]. Soil properties (total salt, total nitrogen, and soil organic matter) were found to be the most important factors affecting the abundance and diversity of bacterial phyla, which is consistent with the results of other

large-scale studies [59–61]. In this study, the soil moisture content, organic matter, and total nitrogen are significant physicochemical factors influencing crop plantation in the yellow river irrigation area.

Soil moisture content is the primary factor affecting the composition of bacterial communities. If the soil moisture content is reduced, it can inhibit aerobic microbial growth and promote the growth of certain anaerobic microorganisms, such as Chloroflexi, which are denitrifying bacteria [62]. The soil water content also limits the decomposition of soil organic matter. Soil pH can serve as a precise predictor of bacterial community composition [10, 18, 63]. All bacterial species can grow well in acidic pH values. Previous studies have shown that soil pH significantly affects the abundance of Actinobacteria, Gemmatimonadetes, Acidobacteriota, and Chloroflexi [62, 64]. The importance of soil pH in influencing bacterial structure and composition has been demonstrated in various studies conducted on different agricultural soils [65, 66]. Alterations in soil pH can potentially affect the nitrogen content in the soil because the initial transformation of nitrogen in the soil-plant system is affected by the concentration of ammonium and nitrate nitrogen. Organic nitrogen mineralization alters the pH value by depleting $H^+$ during ammonification or releasing $H^+$ during nitrification [67–69], indicating the important role of pH in influencing the bacterial community. In our study, the pH was alkaline and exerted lesser influence on bacterial communities; however, an obvious correlation exists among soil moisture, organic matter, total nitrogen, and the abundance of Patescibacteria, Planctomycetes, Gemmatimonadetes, and Acidobacteriota, indicating that changes in soil organic matter and nitrogen content are closely related to variations in bacterial communities during cultivation. The findings of our study indicate that soil bacteria play a major role in mediating nitrogen fixation, nitrification, denitrification, and ammonification [70].

## Conclusions

The bacterial community composition in the soil from eight irrigation areas in northwestern China was significantly different. Bacterial communities tend to be similar in soils with similar physicochemical properties. Soil moisture was positively correlated with the abundance of Patescibacteria and negatively correlated with the abundance of Planctomycetes, Gemmatimonadetes, and Acidobacteriota. Soil organic matter and pH value were positively correlated with the abundance of Planctomycetes. The total salt content was positively correlated with Bacteroidetes abundance. Soil moisture, organic matter content, and total nitrogen were the primary factors that influenced the bacterial community composition in the Yellow River irrigation area. The results of this study will help elucidate the functional contributions of soil bacterial communities to the soil ecosystem and predict the potential function of soil quality in the future, which has gained importance owing to continuous climate change and increasing land-use intensity.

## Supporting information

**S1 File. Physical and chemical property data.**
(XLSX)

**S2 File. Composition of soil bacterial community at phylum level.**
(XLS)

**S3 File. Composition of soil bacterial community at class level.**
(XLS)

**S4 File. α diversity comparison of soil bacteria data.**
(XLS)

**S5 File. NMDS analysis data.**
(XLS)

**S6 File. Correlation analysis between phylum bacteria and physicochemical properties.**
(XLS)

## Acknowledgments

We are grateful to thank Guangzhou Genedenovo Biotechnology Co., Ltd. for testing and assisting in microbiome analysis.

## Author Contributions

**Conceptualization:** Jinjun Cai.

**Data curation:** Xia Wu, Zhangjun Wang, Weiqian Li, Gang Chen, Yangyang Bai.

**Investigation:** Xia Wu, Zhangjun Wang, Weiqian Li, Gang Chen, Yangyang Bai.

**Writing – original draft:** Xia Wu.

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
