## [Editor Report · Decision Letter 0]

30 Apr 2024

PONE-D-24-15589Diversity and Community Distribution of Soil Bacterial in the Yellow River Irrigation Area of Ningxia, China

PLOS ONE

Dear Dr. Cai,

Thank you for submitting your manuscript to PLOS ONE. After careful consideration, we feel that it has merit but does not fully meet PLOS ONE’s publication criteria as it currently stands. Therefore, we invite you to submit a revised version of the manuscript that addresses the points raised during the review process.

The data availability for your manuscript must be properly addressed. On the manuscript you wrote as follows: "Data availability statement: The original contributions presented in the study are included in the article. Further inquiries can be directed to the corresponding author."

During submission process, PLOS ONE indicates that "Stating ‘data available on request from the author’ is not sufficient. If your data are only available upon request, select ‘No’ for the first question and explain your exceptional situation in the text box." However, I did not see any explanation for "upon request" stipulation of your data availability.

PLOS ONE criterion states that "Authors must follow standards and practice for data deposition in publicly available resources including those created for gene sequences, microarray expression, structural studies, and similar kinds of data. Failure to comply with community standards may result in rejection."

We look forward to receiving your revised manuscript.

Kind regards,

Rina Bagsic Opulencia, PhD

Academic Editor

PLOS ONE

“The authors wish to express their sincere thanks to the Ningxia Hui Autonomous Region Sci-Tech Innovation Demonstration Program of High-Quality Agricultural Development and Ecological Conservation (NGSB-2021-11-01),and  the Ningxia Hui Autonomous Region Key R&D Project(2023BEG02042) . The authors are grateful to thank Guangzhou Genedenovo Biotechnology Co., Ltd. for testing and assisting in microbiome analysis.”

5. We note that your Data Availability Statement is currently as follows: [All relevant data are within the manuscript and its Supporting Information files.]

6. We note that Figure 1 in your submission contain [map/satellite] images which may be copyrighted. All PLOS content is published under the Creative Commons Attribution License (CC BY 4.0), which means that the manuscript, images, and Supporting Information files will be freely available online, and any third party is permitted to access, download, copy, distribute, and use these materials in any way, even commercially, with proper attribution. For these reasons, we cannot publish previously copyrighted maps or satellite images created using proprietary data, such as Google software (Google Maps, Street View, and Earth). For more information, see our copyright guidelines: http://journals.plos.org/plosone/s/licenses-and-copyright.

---

## [Author Response · Author response to Decision Letter 0]

7 May 2024

Revision notes are given as follows:

1.Revise the manuscript according to the PLOS ONE's style requirements.

2.The data used for manuscript collection is obtained through on-site investigation and sampling by the project team in the research area, soil physical and chemical properties, and testing, without the need for a permit.

3.The R code for the drawing will be uploaded after acceptance.

4.The Ningxia Hui Autonomous Region Agricultural High Quality Development and Ecological Protection Technology Innovation Demonstration Project (NGSB-2021-11-01) and the Ningxia Hui Autonomous Region Key R&D Project (2023BEG02042) jointly provide funding support for research work, including soil sampling investigation, soil sample testing, data collection and analysis, decision to publish, preparation of the manuscript. The Guangzhou Genedenovo Biotechnology Co., Ltd assists in soil bacterial testing and analysis.

5.All data used in the submission has been listed with file names in the revised manuscript and has been submitted.

6.Delete Figure 1 from the manuscript, which includes a map. Deleting Figure 1 does not affect the description of the sampling area in the manuscript, as Table 1 already lists the specific information of the sampling area.

---

## [Decision Letter · Decision Letter 1]

13 Sep 2024

Diversity and Community Distribution of Soil Bacterial in the Yellow River Irrigation Area of Ningxia, China

PONE-D-24-15589R1

Dear Dr. Cai,

We’re pleased to inform you that your manuscript has been judged scientifically suitable for publication and will be formally accepted for publication once it meets all outstanding technical requirements.

Kind regards,

Rina Bagsic Opulencia, PhD

Academic Editor

PLOS ONE

Additional Editor Comments (optional):

Reviewers' comments:

Reviewer's Responses to Questions

**Comments to the Author**

1. If the authors have adequately addressed your comments raised in a previous round of review and you feel that this manuscript is now acceptable for publication, you may indicate that here to bypass the “Comments to the Author” section, enter your conflict of interest statement in the “Confidential to Editor” section, and submit your "Accept" recommendation.

Reviewer #1: (No Response)

Reviewer #2: All comments have been addressed

2. Is the manuscript technically sound, and do the data support the conclusions?

Reviewer #1: Yes

Reviewer #2: Yes

3. Has the statistical analysis been performed appropriately and rigorously? 

Reviewer #1: Yes

Reviewer #2: Yes

4. Have the authors made all data underlying the findings in their manuscript fully available?

Reviewer #1: Yes

Reviewer #2: Yes

5. Is the manuscript presented in an intelligible fashion and written in standard English?

Reviewer #1: Yes

Reviewer #2: Yes

6. Review Comments to the Author

Reviewer #1: In line 103, change "instructions" to "manufacturer's protocol".

In line 104, chsange "detect" to "determine".

In line 108, indicate the threshold value applied for taxonomic assignation.

In lines 113, 115 and 118, specify the R final version used for all conducted analysis, since you specified two versions (4.3.1 and 4.2.2).

In line 116, change "among" to "between".

In line 117, change "test" to "post hoc test".

In line 136, change "OUT" to "OTUs".

In line 143, in the range values the lowest value is missing.

In line 161, remove "plots".

In line 163, change "The Coverage index" to "The Good's coverage index".

In line 164, change "with good sample coverage" to "showing homogeneous sequencing coverage across samples".

In line 165, change "plots" to "analysis".

In line 200, change "MiSeq technique" to "microbial diversity analysis".

In lines 217, 225, 226 and 227, turn "Proteobacteria" into italics.

In line 286, turn "Planctomycetes" into italics.

Reviewer #2: Dear editor,

I think the authors have made the required revisions and it can be acceptable now.

Yours sincerely

7. PLOS authors have the option to publish the peer review history of their article (what does this mean?). If published, this will include your full peer review and any attached files.

Reviewer #1: **Yes: **Aarón Barraza

Reviewer #2: No

---

## [Editor Report · Acceptance letter]

19 Sep 2024

PONE-D-24-15589R1 

PLOS ONE

Dear Dr. Cai, 

I'm pleased to inform you that your manuscript has been deemed suitable for publication in PLOS ONE. Congratulations! Your manuscript is now being handed over to our production team.

Kind regards, 

on behalf of

Dr. Rina Bagsic Opulencia 

Academic Editor

PLOS ONE